



# ML-SWAN-v1: a hybrid machine learning framework for the prediction of daily surface water nutrient concentrations

Benya Wang[1,2], Matthew R. Hipsey[2,3], Carolyn Oldham[1,2]

[1]Department of Civil, Mining and Environmental Engineering, The University of Western Australia, 35 Stirling Highway, Crawley 6009, Australia
[2]Co-operative Research Centre for Water Sensitive Cities, Clayton, Australia
[3]UWA School of Agriculture and Environment, The University of Western Australia, 35 Stirling Highway, Crawley 6009, Australia

*Correspondence to*: Carolyn Oldham (carolyn.oldham@uwa.edu.au)

**Abstract.** Nutrient data from catchments discharging to receiving waters are necessary to monitor and manage water quality, however, they are often sparse in time and space and have non-linear responses to environmental factors, making it difficult to systematically analyse long- and short-term trends and undertake nutrient budgets. To address these challenges, we developed a hybrid machine learning (ML) framework that first separated baseflow and quickflow from total flow, and then generated data for missing nutrient species, using relationships with hydrological data, rainfall, and temporal data. The generated nutrient data were then included as additional variables in a final simulation of tributary water quality. Hybrid random forest (RF) and gradient boosting machines (GBM) models were employed and their performance compared with a linear model, a multivariate weighted regression model and stand-alone RF and GBM models that did not pre-generate nutrient data. The six models were used to predict TN, TP, NH3, dissolved organic carbon (DOC), dissolved organic nitrogen (DON), and filterable reactive phosphorus (FRP) discharged from two study sites in Western Australia: Ellen Brook (small and ephemeral) and the Murray River (large and perennial). Our results showed that the hybrid RF and GBM models had significantly higher accuracy and lower prediction uncertainty for almost all nutrient species across the two sites. We demonstrated that the hybrid model provides a flexible method to combine data of varied resolution and quality, and is accurate for the prediction of responses of surface water nutrient concentrations to hydrologic variability.

## 1 Introduction

Surface water nutrient concentrations have been significantly increased by human activities (Forio et al., 2015) due to urbanisation, waste discharges and agricultural intensification (Liu et al., 2012; Kaiser et al., 2013; Li et al., 2013). Increased nutrient concentrations and loads in streams alter the biogeochemical functioning and biological community structure in receiving estuaries (Jickells et al., 2014; Staehr et al., 2017), leading to increased incidence of harmful algal blooms (Domingues et al., 2011), anoxia and hypoxia (Li et al., 2016; Testa et al., 2017), and reduced water availability (Heathwaite,



2010). Analysis of tributary water quality data over time is therefore essential to compute incoming nutrient loads, support policy, and plan remediation measures.

Water quality data, however, often have constraints that make it challenging to analyse long- and short-term trends. Firstly,
water quality data often have non-linear responses to environmental factors and show high-order interaction effects between different environmental variables. Moreover, nutrients can derive from different sources (point or non-point) in the landscape and are transported to receiving waters through different water pathways subject to varied catchment hydrological conditions and human intervention (Hirsch et al., 2010; Lloyd et al., 2014). Additionally, tributary nutrient datasets often are sparse in both space and time, due to the high cost of fieldwork and chemical analysis (Lamsal et al., 2006; Forio et al., 2015). Historical
and current water quality monitoring programmes often use low-frequency sampling regimes on a weekly to monthly basis (Halliday et al., 2012). When monthly averaged concentrations are used, calculated nutrient loads to receiving environments such as lakes or estuaries may be poorly estimated (Cozzi and Giani, 2011), with high variability in the estimated loads (Jordan and Cassidy, 2011). It is also common to have patchy availability of nutrient species data across a study area, and combining datasets from different projects and analytical laboratories make the analysis of long-term trends fraught with uncertainty. For
instance, total nitrogen (TN) and total phosphorus (TP) concentrations within catchment outflows, may have been monitored for decades, while dissolved organic nitrogen (DON) and dissolved organic carbon (DOC) concentrations may have only been monitored recently, with the increasing recognition of their ecological importance (Górniak et al., 2002; Petrone et al., 2009; Erlandsson et al., 2011). Given the hydrochemical correlation between different nutrient species and high analytical cost, there are benefits in extracting maximum information from all available nutrient data, particularly relating to changes in water quality
over time (Hirsch et al., 2010). In summary, while high-quality nutrient data from tributaries are typically required as input to water quality modelling of receiving waters, the reliability and accuracy of trend analysis of tributary data are frequently restricted by data non-linearity, limited sample size, and variable nutrient availability.

Various models for constructing tributary water quality data have been developed. For example, linear models (LM) and
generalised linear models (GLM) that use correlations between concentration (C) and flow (Q), have long played a central role in stream water quality analysis (Cohn et al., 1989; Chanat et al., 2002). Some multivariate regression models have been applied to analyse the long-term trend (Li et al., 2007; Tao et al., 2010; Greening et al., 2014) and seasonal patterns (Giblin et al., 2010; Chen et al., 2012) of surface water nutrients. For example, a weighted regression on time, discharge, and season (WRTDS) was introduced by Hirsch et al. (2010) and has been applied to a number of different water quality studies (Green
et al., 2014; Zhang et al., 2016a, 2016b, 2016c).

Meanwhile, data-driven machine learning (ML) methods are increasingly being applied to quantify relationships between soil, water and environmental landscape attributes (Lintern et al., 2018; Wang et al., 2018; Guo et al., 2019). For instance, random forest (RF), a widely used ML method, was used to model the spatial and seasonal variability of nitrate concentrations in





streams (Álvarez-Cabria et al., 2016). Gradient boosting machines (GBM) were used to quantify relationships between land-use gradients and the structure and function of stream ecology (Clapcott et al., 2012). In contrast to process-based conceptual models, ML methods simulate relationships purely from the data (Maier et al., 2014) and have the ability to incorporate different types of variables (e.g. numerical or categorised variables); this is particularly suitable for systems with complex variable interactions and non-linear response functions (Povak et al., 2014).


While both process-based and ML models can manage non-linear interactions and be used to explore long-term trends, they both have difficulty in fully extracting important hydrochemical information embedded in nutrient data. Hybrid methods have been proposed for flow forecasting, to enhance the performance of ML models by first using intermediate models to generate additional variables, which are then used for subsequent modelling. For instance, a neural network model is first applied to

reconstruct surface ocean partial pressure of carbon dioxide (pCO2) climatology, which is used as a input into another neural network to predict pCO2 anomalies with other features (Denvil-Sommer et al., 2019). Similarly, Noori and Kalin (2016) used the soil and water assessment tool (SWAT) to generate baseflow and stormflow, which were then used as inputs to an ANN model to improve daily flow prediction. Both studies used hybrid models to demonstrate that pre-generated variables provided additional information that was crucial to achieving higher prediction accuracy, compared with stand-alone ANN models.


Stream flow integrates water from multiple pathways resulting in a distribution of residence times. Stream nutrients are the product of over-lapping historical inputs and reaction rates, which are spatially distributed and temporally weighted within the catchment (Abbott et al., 2016). Therefore, it is beneficial to understand nutrient transport pathways from the source to receiving waters, to analyse the long- and short-term trends of stream nutrient data; this knowledge will improve management

strategies to reduce nutrient transport (Tesoriero et al., 2009; Mellander et al., 2012). In the analysis of the streamflow hydrograph, separating baseflow (the long-term delayed flow from storage) and quickflow (the short-term response to a rainfall event) from total flow is a well-established strategy to better understand transport pathways (Tesoriero et al., 2009). To utilise all available nutrient data and assess the impact of different transport pathways on stream nutrient concentrations, we developed a hybrid machine learning framework for surface water nutrient concentrations (ML-SWAN) that first separated baseflow and

quickflow from total flow, then built intermediate models to generate missing nutrient species within the total nutrient pool, using relationships with baseflow, quickflow, rainfall, and temporal data. The generated nutrient data were included as additional variables for a final ML prediction. RF and GBM were employed and their performance compared in stand-alone mode and as a hybrid method.

This study aimed to compare model performance for nutrient concentration prediction, to generate accurate daily nutrient data, to assess the impacts of different water transport pathways on surface water nutrient concentrations, and to present a feasible framework for the application of the hybrid method for surface water nutrient prediction. It was hypothesised that the hybrid RF and hybrid GBM, which used pre-generated daily nutrient concentrations and the separated baseflow and quickflow as





additional auxiliary inputs, would take advantage of the complementary strengths of hydrochemical and hydrological
relationships to provide the most accurate and reliable nutrient predictions. To test this hypothesis, the hybrid RF and hybrid
GBM were compared to a linear model, a multivariate weighted regression model (WRTDS), and stand-alone RF and GBM
models, for the prediction of TN, TP, ammonia ($NH_3$), DOC, DON, and filterable reactive phosphorus (FRP) concentrations,
at two different sites under varied hydrological conditions.

## 2 Model overview

Our modelling goal in this study was to minimise the sum of the overall loss function between the predicted nutrient
concentrations and measured nutrient concentrations.

$$\sum_i L(y_i, F(X_i)) \qquad (1)$$

where $L$ is a loss function (e.g., squared error), $y_i$ are measured values, $X_i$ are relevant variables, $F$ is any approximation model,
and $F(X_i)$ or $\hat{y}_i$ is the model-predicted value at $X_i$. The descriptions of different approximation models are described in the
following sections.

### 2.1 Linear model and WRTDS model

Linear models (LM) are the most commonly used tool to describe concentration-discharge (C-Q) relationships (Hirsch et al.,
2010). Typically, a *log* transformation is often applied to *C* and *Q* data (Crowder et al., 2007; Meybeck & Moatar, 2012;
Herndon et al., 2015), with the linear model then described as:

$$\log(C) = \beta_0 + \beta_1 \log(Q) \qquad (2)$$

where *C* is nutrient concentration and *Q* is total flow. In this study, the linear model was used as a benchmark for other models.
The fitted slope $\beta_0$ can represent the base nutrient concentration in a stream, while $\beta_1$ can describe relationships between
hydrological and biogeochemical data. The WRTDS model was also used (Hirsch et al., 2010) and can be described as:

$$\log(C) = \beta_0 + \beta_1 \log(Q) + \beta_2 JD + \beta_3 \sin(JD) + \beta_4 \cos(JD) + \varepsilon \qquad (3)$$

where *JD* is the Julian day, $\varepsilon$ is unexplained variation. $\beta_2 JD$ is used to represent the long-term trend from year to year, while
$\beta_3 Cos(JD)$ and $\beta_4 Sin(JD)$ are used to describe the seasonal variation of stream nutrient concentrations. To calculate the Julian
Day for use in Eq 3, days since 01/01/1970 was first calculated and then multiplied by $2\pi$. WRTDS advances the simpler linear
model in two aspects. Firstly, the additional components in the equation allow consideration of seasonal and long-term patterns
and make the WRTDS model more able to describe stream nutrient concentrations across the year. Secondly, unlike the linear
model whose parameters are constant in time, WRTDS adjusts the parameters in a gradual manner throughout *Q*, *JD* space.
This is accomplished by applying a weighted regression for the estimation of *log(C)*, where the weights on each observation





are based on three distances between the observation ($Q_o$, $JD_o$) and the estimation point ($Q_i$, $JD_i$) which are 1) the time distance between $JD_o$ and $JD_i$, 2) the seasonal distance between the time of year at $JD_o$ and the time of year at $JD_i$, and 3) the discharge distance between $log(Q_o)$ and $log(Q_i)$ (Hirsch et al., 2010; Green et al., 2014). Thus, *log(C)* is considered to be locally linearly related to *log(Q)*, *JD*, *Sin(JD)*, and *Cos(JD)*.

### 2.2 Random forest and gradient boosting machines

Random forest (RF) and gradient boosting machines (GBM) are ensemble models that combine multiple base learners inside the model to improve the prediction performance (Ishwaran and Kogalur, 2010; Singh et al., 2014). The ensemble methods are the main difference between RF and GBM. In RF, bootstrap aggregating is used to resample the original dataset with replacement. Hence, datasets with partial data are generated and then used to build individual base learners. Unlike bootstrap aggregating, GBM iteratively generates a sequence of base learners, where each successive base learner is built for residual

prediction of the preceding base learner (Friedman, 2001, 2002). The probability with which data points are selected for the next training set is not constant and equal for all data points. The selection probability increases for data points that have been mis-estimated in the previous iteration; data points that are difficult to classify would receive higher selection probabilities than easily classified data points (Yang et al., 2010; Erdal & Karakurt, 2013).

For RF and GBM, the most commonly used base learner is a classification and regression tree (CART). A CART model is built to split the dataset into different nodes (Breiman et al., 1984), $\left\{ X_1, x_i^a < v \right\}$ and $\left\{ X_2, x_j^a \geq v \right\}$ for numeric variables or $\left\{ X_1, x_i^d = c \right\}$ and $\left\{ X_2, x_j^d \neq c \right\}$ for categorised variables, where $i$ and $j$ are the sample indices, $a$ is a numerical variable, $v$ is one of the values of $a$ variable, $d$ is a categorised variable, and $c$ is one of the values of $d$ variable. To split the dataset at $a$ or $d$, the sum of least-square error of the two nodes are calculated for a regression task as:


$$error = \sum_{l=0}^{L}(y_l - \overline{y}_L)^2 + \sum_{r=0}^{R}(y_r - \overline{y}_R)^2 \qquad (4)$$

where $y_l$ and $y_r$ are observations in two split nodes, and $\overline{y_L}$ and $\overline{y_R}$ are the average $y$ in that node. The split is chosen among all candidate variables and values to minimise this error. This splitting process is applied from the root to the terminal node,

which creates a tree structure for the model (Erdal and Karakurt, 2013). A CART can be used both for classification and regression problems due to this tree-structure (Coops et al., 2011). However, a single CART can sometimes over-simplify variable interactions and may lead to low prediction performance (McBratney et al., 2000; Cutler et al., 2007; Coopersmith et al., 2010). This drawback can be overcome by the ensemble method that generates many resampled datasets and creates various CARTs to achieve higher accuracy (Breiman, 2001) and more stable results when facing slight variations in input data

(Martínez-Rojas et al., 2015). New data input is thus evaluated against all trees created in the ensemble model and each tree votes using the main class or the averaged values in the terminal node. The class with the maximum votes will be used for a





classification model, and the averaged predicted value from all trees is used for a regression model (Singh et al., 2014; Belgiu & Drăgu, 2016;). It is found that ensemble methods in RF and GBM can significantly improve the prediction accuracy of CART (Ismail & Mutanga, 2010; Erdal & Karakurt, 2013). GBM models are generally have less decision tree models than
RF.

Compared to LM and WRTDS models, one drawback of RF and GBM, as well as many ML methods in general, is that there is no specific equation in GBM or RF to directly demonstrate model structures. However, GBM and RF do provide the relative importance of each variable, which is based on the empirical improvement in the loss function due to the split on the specific
variable in a tree (Povak et al., 2014; Puissant et al., 2014). The improvement of a certain variable was averaged over all trees, and used as the relative importance of that variable for the final model. This relative importance serves as the key index to understand the model structure of RF and GBM (Makler-Pick et al., 2011).

### 2.3 Baseflow separation

Total flow is commonly conceptualised as including baseflow and quickflow components (Meshgi et al., 2015). Baseflow
separation techniques use the time-series record of streamflow to extract the baseflow and quickflow signatures from the total flow. This can be done by using graphical methods to identify the intersection between baseflow and the rising and falling limbs of the quickflow response (Szilagyi and Parlange, 1998), or filtered methods which process the entire stream hydrograph to derive a baseflow hydrograph (Furey and Gupta, 2001). In this study, the three passes filtered method was applied for baseflow separation; the quickflow was first estimated as described below (Lyne and Hollick, 1979; Nathan and McMahon,
1990), and then baseflow calculated:

$$QF_i = \alpha QF_{i-1} + (Q_i - Q_{i-1})\frac{1+\alpha}{2} \qquad (5)$$

where $QF_i$ is the filtered quickflow for the $i^{th}$ sampling instant, $QF_{i-1}$ is the filtered quickflow for the previous sampling instant to $i$, $\alpha$ is the filter parameter with a value of 0.925 for daily flow as recommended by Nathan and McMahon (1990). Baseflow
is then calculated as $BF = Q - QF$.

### 2.4 Performance evaluation metrics

In this study, the root mean squared error (RMSE) and the Nash–Sutcliffe model efficiency coefficient (MEF) were used to compare model performance. The RMSE is a measure of overall error between the predicted and measured data and returns an error value with the same units as the data, which is given by the following equation:

$$RMSE = \sqrt{\frac{\sum(y_i - \hat{y}_i)^2}{n}} \qquad (6)$$





where $n$ is the number of data samples. RMSE varies from 0 to $+\infty$, and a perfect model would have RMSE of 0. The MEF is a dimensionless ''goodness-of-fit'' measure which can vary from $-\infty$ to 1, with a value of 1 indicating a perfect fit and 0 indicating that the mean of the measured values performs as well as the model. The MEF can be calculated as:

$$MEF = 1 - \frac{\sum(y_i - \hat{y}_i)^2}{\sum(y_i - \overline{y}_i)^2} \qquad (7)$$

where $\overline{y}_i$ is the mean of the measured values. Note that the predicted and measured nutrient values were normalised to [0, 1] in this study to compare model performance across different nutrient species.

### 2.5 Overview of modelling processes

The data points were divided into the training dataset (80%) and the testing dataset (20%). Different models were built and tuned on the training dataset; the testing dataset was saved for the final test. Five-fold cross-validation (CV) was done on the training dataset to tune the model parameters. The performance of all six methods (LM, WRTDS, RF, GBM, hybrid RF, and hybrid GBM) was evaluated on the testing dataset. WRTDS was run through the EGRET (Exploration and Graphics for RivEr Trends) package (Hirsch and De Cicco, 2015) in R to produce daily concentrations for six nutrient species (TP, TN, DON,

DOC, NH$_3$, and FRP). The default settings specified by the user guide (Hirsch and De Cicco, 2015) were used. RF and GBM models were built through the H$_2$O package in R.

To assess the prediction uncertainty of the six models, the divided and tested processes were repeated 30 times (the process within the dashed line in Figure 1) except WRTDS. Leave-one-out cross-validation (LOOCV) was used in WRTDS to predict

daily nutrient concentrations; LOOCV is the default cross-validation method in the EGRET package. In that method, one data point was excluded at a time from the whole dataset, all other data points were used to build the model, and the excluded point was used for testing the model performance. This process was repeated for all data points.

The overall processes of ML-SWAN can be divided into three stage (Figure 1). The first stage was baseflow separation using

the EcoHydRology package (Fuka et al., 2018). The generated baseflow, quickflow, total flow and rainfall were further transformed into lagged data (the averaged values over the previous 3, 7 and 15 days) to capture any short-term impacts of different water pathways and rainfall on stream nutrients. *JD*, *Cos(JD)*, and *Sin(JD)* were also calculated for RF and GBM to include seasonal and long-term impacts. A description of all the used variables is given in Table 1.

The second stage of ML-SWAN was to build intermediate RF and GBM models that generated daily nutrient concentrations. For the intermediate RF and GBM models, only lagged hydrological data (including total flow, baseflow, and quickflow), lagged rainfall, temporal data on the training dataset were used. Nutrients were not used as a predictor in the intermediate model. Note that, in this study TP, TN, DOC, and DON were selected to be generated in the second step. If one nutrient was





considered as the final target, the other three nutrients were used to generate daily data. For instance, daily TP, DOC, and DON

were generated as additional variables to predict TN. In that case, the missing TP, DOC, and DON were generated by the intermediate model for the training dataset and the testing dataset. Daily TN, TP, DOC, and DON data were generated and used for the final predictions. These nutrients were selected since they may be generated from similar sources, or are important components of the total nutrient load. For instance, DOC and DON may both be generated from DOM (Seitzinger et al., 2002; Bernal et al., 2005; Filep & Rékási, 2011). In the catchments studied here, DON can be a dominant component of TN (Nice et

al., 2009; Petrone, 2010; Bourke et al., 2015). The selection of DOC and DON for pre-generation may not necessarily be appropriate for other catchments. The selection of nutrients for pre-generation depends on data availability in the dataset. The use of different species of the same nutrients (N or P) can generally improve model performance.

The third stage of ML-SWAN built an additional hybrid model using the training data, which has generated nutrient data by

the intermediate models, lagged hydrological data, lagged rainfall data and temporal data. Note that at this stage, the only difference between stand-alone ML and hybrid ML methods was that stand-alone ML did not use pre-generated daily nutrient data.

**3. Site overview**

To test the generalisability of the hybrid framework, two sites in Western Australia (Ellen Brook and Murray River) were

selected as study areas. Ellen Brook and Murray River are key tributaries for Swan-Canning Estuary and Peel-Harvey Estuary (Figure 2), respectively, and have different hydrological conditions. The Swan-Canning Estuary is located adjacent to the Perth metropolitan area, with an area of approximately 40 km$^2$. The catchment comprises 30 catchments which drain approximately 2090 km$^2$ (Kelsey et al., 2010). Ellen Brook is the largest sub-catchment in the Swan-Canning catchment, comprising 34% (716 km$^2$) of the total catchment area. Ellen Brook is an ephemeral river with no flow recorded during summer and early

autumn months (Table 2). The dominant land use in Ellen Brook is agricultural and grazing land. Ellen Brook is one of the highest contributors of TN and TP to the Swan-Canning Estuary (Swan River Trust, 2009). Bassendean sands and duplex Yanga (sand over clay) soils dominate the Ellen Brook catchment. Bassendean sands have very low phosphorus retention indices (PRI), while Yanga soils have low PRI in their upper horizon and become waterlogged in winter, promoting the release of retained nutrients to the stream (Kelsey et al., 2010).


The Peel–Harvey Estuary is located approximately 75 km south of the Swan-Canning Estuary and the Serpentine, Murray and Harvey Rivers drain into the estuary (Figure 2). The total catchment area of the estuary is approximately 11930 km$^2$. The Murray River catchment is dominated by deep grey sands, loams clay and peats (Ruibal-Conti et al., 2013), agricultural land use and natural reserves, and it contributes about 40% of annual TN loads and 7% of annual TP loads to the estuary (Kelsey

et al., 2011).





Both Swan-Canning Estuary and Peel-Harvey Estuary experience a Mediterranean climate with cool, wet winters (June–August) and hot, dry summers (December–March). The long-term average annual rainfall varies from 1300 mm on the coast, to 800 mm in the southeast of the catchment area (1975–2009, Bureau of Meteorology station), and about 90% of the rain falls

between April and October. Sample size and the first measurement year of six nutrients species are listed for the two study sites in Table 3. TN, TP, NH₃, and FRP have been monitored for decades, while DOC and DON have only been measured in recent years, with limited sample size. Several historical nutrient datasets were combined but significant changes occurred in water sampling devices and analytical instrumentation over the past decades. These changes can increase the complexity of nutrient data. For instance, auto-samplers sampled any time regardless of weather conditions (e.g., during the rainfall) while

grab samples were typically collected under fine weather conditions due to safety concerns.

## 4. Results

### 4.1 Comparison of prediction accuracy between six methods

Overall, the scaled RMSE reduced from LM, WRTDS, stand-alone ML, and hybrid ML for all nutrients except NH₃, and the opposite pattern was found for MEF in both Ellen Brook and Murray River (Figure 3). The linear model had the worst

performance: the scaled RMSE was significantly higher and MEF was significantly lower than the other models, for all six nutrients and across both sites. WRTDS generally had higher RMSE and lower MEF than the stand-alone ML, although it achieved similar results to stand-alone ML for FRP and NH₃ at both sites. LOOCV was used in WRTDS and only one set of results were generated, compared to 30 RMSE and MEF values for other methods. This results in a shortened line for WRTDS in Figure 3, instead of the inter-quartile ranges (IQR = 75$^{th}$ percentile - 25$^{th}$ percentile) presented for the other methods. LOOCV

can sometimes over-estimate the model performance as only one sample was tested at a time; in contrast, 20% of the independent testing data were tested in the other five models. LOOCV can also have a higher variance than other CV methods (Li, 2016).

Stand-alone ML achieved results that placed it between WRTDS and hybrid ML. Stand-alone GBM achieved the highest

accuracy for NH₃ prediction in Murray River. Hybrid RF and hybrid GBM had the lowest RSME and highest MEF for all nutrients except NH₃, in Ellen Brook and Murray River (Figure 3). Compared to the stand-alone ML, the hybrid ML also had much lower prediction uncertainty, in that the RMSE and MEF had narrower IQR than that of the stand-alone ML, especially for DON and FRP prediction in Ellen Brook and DOC prediction in Murray River. The use of pre-generated daily nutrient data was the only difference between hybrid ML and stand-alone ML. This means that the generated nutrients provided

additional information for the hybrid model that allowed more stable results. Interestingly, while the hybrid ML had better performance than the stand-alone ML, there was no significant difference in performance between the hybrid RF and hybrid GBM, though they showed differences between different nutrient species. For instance, hybrid RF achieved slightly better





performance for DOC in Ellen Brook, while hybrid GBM had lower RMSE for DOC in Murray River. There was no significant performance difference between stand-alone RF and GBM.


In summary, the hybrid ML had the best performance amongst the six methods, followed by stand-alone RF and GBM. WRTDS was better than the linear model but could only achieve results similar to stand-alone RF and GBM for NH$_3$ prediction in Ellen Brook, and for NH$_3$ and FRP prediction in Murray River.

### 4.2 Generated daily TN in Ellen Brook

These six models were compared in their ability to generate daily TN in Ellen Brook from 01/01/1989 to 16/07/2018 (Figure 4). Note that all data points (not just the 80% training dataset) were used to generate daily TN. Similar results to those presented in the above section, were found for the generated daily TN in the Murray River, and for TP in both Ellen Brook and the Murray River (results in Appendix A).

The LM performed very poorly for TN prediction; low concentration samples (TN < 1.9 mg/L) were all under-estimated, and some extremely high concentrations were incorrectly generated due to the high flow (Figure 4-a). LM only used total flow to predict nutrient concentrations while other important hydrological processes were ignored. Thus the over-simplified LM had high errors in nutrient prediction (Figure 3), and this method might be more suitable for solutes that are not substantially bioactive (e.g., $SiO_2$, $Ca^{2+}$, $Mg^{2+}$, $Cl^-$) (Stallard and Murphy, 2014). The WRTDS captured some seasonal patterns of TN (from

2008 to 2018) but still had problems predicting TN between 1989 to 1996; some extremely high values were generated, and TN < 1.0 mg/L were over-estimated (Figure 4-b). Stand-alone ML and hybrid ML generated similar daily TN data but varied in the detail. These models successfully captured the low concentration data and the seasonal pattern of TN. The RF and hybrid RF both under-estimated a few high concentration data (TN > 4.0 mg/L), compared to GBM and hybrid GBM, although hybrid RF still showed better performance than RF. For instance, high concentration data in 2007 and again from 2014 to 2017 were

successfully predicted by hybrid RF but under-estimated by RF. Compared to stand-alone GBM, the hybrid GBM achieved lower errors for high concentration data.

Apart from the better performance for high concentration data, another difference between stand-alone ML and hybrid ML was that the long-term trend in TN was consistent in stand-alone ML, but this trend fluctuated in hybrid ML. For instance,

hybrid GBM results fluctuated from 1989 to 1999 and then showed an increasing long-term trend from 2000 to 2018, in addition to the seasonal fluctuation. This suggests that the generated nutrient data provided additional information that allowed the hybrid ML to capture long-term trends; this information was not included in the temporal data, but existed in the generated nutrient data.



The distribution of the TN data generated by the six models was compared to the distribution of the measured TN data (Figure 5). Similar to the results shown in Figure 4, hybrid GBM had the most similar distribution to the measured TN data. Only a few high concentration data (TN> 4.0 mg/L) were under-estimated. Hybrid RF also achieved a distribution similar to the measured data, but several high concentration data were under-estimated. GBM showed an improved distribution compared to RF, while WRTDS generated some extremely high data, as shown in Figure 4-b. The linear model incorrectly predicted

most of the TN data. The results in both Figure 4 and Figure 5 show that hybrid GBM achieved the best simulated daily TN data, followed by hybrid RF, stand-alone GBM, and RF. WRTDS and LM generated large biases in TN prediction.

The hybrid ML models predicted most of the extreme concentrations (Figure 4 and Figure 5) and only a few points were under-predicted. The limited number of extreme data and the model structure that tried to balance the overall trend prediction with

extreme data prediction can cause the under-prediction. For example, higher weights can be set up for extreme data during the model training process to force model to over-predict the value for extreme concentrations, which may reduce the accuracy for overall trend prediction. In this study, our target is to understand the long-term nutrient trend. Therefore, we did not use this technique during the model training process.

### 4.3 Comparison of variable importance in hybrid GBM for TN prediction

The daily data generated by the hybrid GBM showed lower RMSE and better distribution than stand-alone ML, WRTDS, and LM (Figure 4 and Figure 5). Compared to LM, WRTDS, and simple CART models, one drawback of RF and GBM, as well as many ML methods in general, is that there is no specific equation in GBM or RF to directly demonstrate model structures. However, GBM and RF do provide the relative importance of each variable, which is based on the empirical improvement in the loss function due to the split on the specific variable in a tree (Povak et al., 2014; Puissant et al., 2014). The improvement

of a certain variable was averaged over all trees as the relative importance for the final model. This relative importance serves as the key index to understanding the model structure of RF and GBM (Makler-Pick et al., 2011).

The variable importance for TN prediction by hybrid GBM in Ellen Brook and Murray River is presented in Figure 6. The variable importance in the intermediate models is also included, and the length of coloured sections represents the importance

of those variables in the hybrid GBM or intermediate GBM. The importance was scaled according to the most important variable. The generated DON and TP ranked as the first two critical variables in Ellen Brook, while all three generated nutrients were listed as the most important variables in Murray River. This suggests that the generated nutrients do provide critical information to the model and improve model performance. The quickflow was most important for the generated DON and TP, as well as the TN itself in Ellen Brook. The impacts of quickflow decreased, and baseflow, temporal data and rainfall data

become more important for TN prediction in Murray River. This difference in variable importance reflects different catchment characteristics across the two sites, and therefore different hydrological and hydrochemical processes controlling TN concentrations. The total flow was not of high importance in either site, which suggests that baseflow or quickflow had more





impact on surface water TN. Moreover, TN concentrations were affected by more variables in Murray River than in Ellen Brook.

**5. Discussion**

**5.1 Different sources of TN in Ellen Brook and Murray River**

Hydrological conditions, specific sub-catchment characteristics and the chemical properties of nutrients can all impact surface water nutrient concentrations (Barron et al., 2009; Moatar et al., 2016), nutrient partitioning (Ruibal-Conti et al., 2013), and nutrient transport (Burt and Pinay, 2005; Tesoriero et al., 2009). TN prediction in Murray River was impacted by more

variables than in Ellen Brook, (Figure 6), suggesting more complex relationships in Murray River.

Quick flow is composed of runoff, interflow and direct precipitation (Brodie and Hostetler, 2005), and was shown to be important for TN prediction in Ellen Brook. Direct precipitation, however, did not have a large impact on TN (the green bars in Figure 6); this suggests that runoff and interflow were important for TN concentrations. Baseflow can account for (on

average) 53% of annual stream discharge in Ellen Brook, but baseflow was not of high importance for TN prediction in this study. This may occur due to low TN concentrations in the baseflow (Barron et al., 2009), large areas of low nutrient-retaining sandy soils in Ellen Brook catchment, and high nutrient transport efficiency in quickflow and first flush. Mellander et al. (2012) quantified nutrient transport pathways in agricultural catchments and found that quickflow were only 2~8% of total flow, but it can transport up to 50% of TP. Gunaratne et al, (2017) found that the seasonal first flush were only 30% of runoff volume

but contained 40~70% of the nutrient load.

Note that the median TN in Ellen Brook (2.1 mg/L) is significantly higher than that in Murray River (0.67 mg/L) which can be explained to some extent by the large area of grazing lands in Ellen Brook. Previous investigations in south-eastern Australia (Adams et al., 2014), New Zealand (Davies-Colley et al., 2004), and north-western Europe (Conroy et al., 2016) all suggested

that livestock can increase TN discharge to the receiving water bodies. Most of the piggeries and poultry farms in the Swan-Canning catchment are located in Ellen Brook catchment (Kelsey et al., 2010), which has the highest TN and TP discharge loads. Thus the large grazing areas, piggeries and poultry farms, and low nutrient-retaining sandy soils may explain the importance of quickflow for TN prediction and high TN concentrations in Ellen Brook.

Baseflow is derived from groundwater discharge to streams and the slow drainage of water stored in local wetlands (Kelsey et al., 2010). Baseflow is highlighted as an important variable for TN prediction in Murray River. Murray River catchment has large areas with high nutrient-retaining soils (high PRI) (Kelsey et al., 2011) and relative low TN concentrations, and it is likely that groundwater makes significant contributions to TN in Murray River. Previous investigations in this study area found that DON was the dominant form of TN in both surface water and groundwater (Nice et al., 2009; Petrone, 2010; Bourke et





al., 2015), and we assumed that groundwater TN contributed to surface water TN in the form of DON. Ruibal-Conti et al. (2014) previously found that variability in TN is strongly associated with variability in flows in Murray River. Our results extend this finding, in that both baseflow and quickflow likely impact TN in the river.

It is noted that temporal data including *Sin(JD)*, and *Cos(JD)* showed significantly higher importance in Murray River. This
indicates stronger seasonal TN signals in Murray River compared to Ellen Brook; this finding is supported by the generated daily TN data for Murray River (see results in Appendix A). Natural reserves occupy large areas of the Murray River catchment, and this may increase seasonal signals. Additionally, the lagged quickflow, baseflow, and rainfall were generated (for the previous 3, 7 and 15 days) but only the lagged 15-day baseflow and quickflow were ranked as important variables for both Ellen Brook and Murray River. This suggests that a time-scale of nutrient transport in the sub-catchments, and likely
reflects soil permeability and geology; long hydrochemical recessions from storm events may prolong their impact on the ecological status of receiving rivers (Mellander et al., 2012).

### 5.2 Can we improve our understanding of historical nutrient conditions using a contemporary data?

The generated nutrient data provided additional information to enhance the hybrid model performance (Figure 3 and Figure 5). To assess the individual impact of a generated nutrient, we did a simple test that gradually added generated nutrient data
into the hybrid GBM and evaluated RMSE and MEF for TN prediction. This process was repeated 30 times and the results are presented in Figure 7.

The RMSE significantly decreased when generated TP was added as an additional variable. DOC and DON only have 297 and 129 data, respectively, and were only measured in recent years, while TP has more than a thousand data and has been measured
since 1990 (Table 3). However, DOC and DON could still improve model performance (Figure 7), and the generated DON was ranked as the most important variable across both sites (Figure 6). The medium RMSE slightly decreased when both generated DOC and DON were added. Moreover, the generated DOC and DON also reduced the model uncertainty, such that the IQR ranges became narrower than model results without the generated nutrients.

Our results suggest that the recent DON and DOC data improved understanding of historical TN. It is not uncommon to have a similar data structure when several datasets are combined, or new measurements are added to a project. While there were no DON data prior to 2006 in Ellen Brook, daily DON can be generated back to 1990 with the help of generated TN, DOC, TP data; DON had the highest MEF among the six nutrients (Figure 3). This hybrid method provides a feasible process to fully utilise all available nutrient data to accurately fill gaps in either historical or recent nutrient datasets.




### 5.3 A comprehensive comparison of six models

Monitoring, modelling, and forecasting water quality inputs are essential to support the management of the quality of receiving
waters while responding to current anthropogenic stressors (Holguin-Gonzalez et al., 2013; Schnoor, 2014). The performances
of six models were comprehensively compared, in an exploration of historical and contemporary nutrient data across two study
sites. LM had the highest error while stand-alone RF and GBM had similar error. This agrees with previous findings by Erdal
and Karakurt (2013) that RF and GBM models achieved similar correlation coefficients (R) for streamflow forecasting. Ismail
and Mutanga (2010) also reported that RF and GBM increased the R of a single CART by 10.01%, and 9.59%, respectively.

The performance of WRTDS, as well as many conceptual models, is often reliant on a prescribed set of input information,
which can account for variance in nutrient concentrations but may miss some important processes for certain rivers (e.g.,
baseflow in this study). This can compromise the performance of WRTDS for nutrient prediction. Moreover, hydrological and
chemical processes within the systems are typically ignored by many conceptual models, which may exclude important
hydrochemical information. By contrast, some complex conceptual models may include these hydrochemical processes but
are often constrained by insufficient nutrient data to calibrate and validate the models. Some simplifications may be made to
account for lack of data, but the simplifications may often weaken model performance. The hybrid framework presented in
this study has overcome the challenge caused by data paucity, by building intermediate models to generate missing nutrient
data, and then using this additional hydrochemical information to improve final model performance.

The hybrid models developed in this study were able to take advantage of the complementary strengths of both hydrochemical
(additionally generated nutrient data) and hydrological (lagged data) information. This was particularly the case for prediction
of high nutrient concentrations, where the hybrid models were shown to outperform the stand-alone RF and GBM, in terms of
accuracy, reliability and value distributions (Figure 4 and Figure 5). Improved accuracy in the hybrid model was achieved by
using intermediate models, although these intermediate models may also have a relatively high error (similar to stand-alone
RF and GBM). However, if the improved model performance is higher than the introduced error, the results are manageable.

A limitation of the hybrid modelling approach, however, is that it requires the time and expertise to develop intermediate
models for generating additional nutrient data. Prior knowledge also plays an important role in identifying the variables for
pre-generation. Some statistical methods (e.g. the correlation test, simple linear model) can be helpful to identify these variables
if there is no clear theoretical or conceptual understanding on which to base selection of the important variables.

In this study, we tested the generalised performance of the hybrid model across six nutrient species and two tributaries. We
also note that nutrients may not always be the critical variables targeted for pre-generation; the pre-generated DOC was ranked



as having low importance for Ellen Brook, and produced only a slight improvement in the performance of the hybrid model

for $NH_3$.

### 5.4 The application of ML methods for hydrological modelling

There were constraints in the nutrient datasets in this research, and similar constraints commonly exist in other study areas. Many nutrient datasets contain important information, but it sometimes can be challenging to directly combine or utilise them. ML methods provide a feasible approach to pre-process these datasets or combine them. In this study, the concentrations of

missing nutrient species were first predicted by the intermediate ML method and then used as inputs for another ML method for final predictions. The pre-generation of missing data and pre-modelling hydrological analysis were critical components of the hybrid model and allowed identification of the impact of different hydrological transport pathways for TN export from the two tributary catchments. The hybrid ML methods were further applied to generate nutrient data for eight tributaries, and the generated data have since been used as inputs to an estuary prediction model, which simulates and forecasts nutrient

concentrations in the previous and next five days in the Swan-Canning Estuary (Huang et al., 2019). The modelling methods and strategies developed in the work presented here, can be easily applied to other study areas. Overall, ML methods provide a flexible and feasible solution to explore the underlying relationships, re-construct spatial and temporal datasets, and combine different models.

### 6. Summary and conclusion

A hybrid machine learning model was developed, and its performance tested on six nutrients and two estuary tributaries and compared with alternative modelling approaches. The hybrid ML model exhibited higher prediction accuracy and lower prediction uncertainty than stand-alone ML, WRTDS, and LM, for almost all nutrients. The pre-generation of missing data and pre-modelling hydrological analysis were critical components of the hybrid model and allowed identification of the impact of different hydrological transport pathways for TN export from the two tributary catchments. The results of this study

demonstrate the advantages of using hybrid models for high temporal resolution nutrient prediction; the results also demonstrate the use of the hybrid model for re-analysis of historical data in the light of contemporary data. Modelling strategies for different modelling targets and dataset structures have also been discussed. The modelling framework presented here can aid others to fully use all available nutrient data to generate accurate nutrient predictions.

*Code and data availability.* The data and the data sources used in this study are cited and explained in the text. Readers can obtain all the necessary data and code from https://github.com/benyawang-uwa/daily-nutrient-prediction





*Author contributions.* Benya Wang, Matthew R. Hipsey, Carolyn Oldham contributed to the development of the methodology and designed the experiments, and Benya Wang carried them out. Benya Wang developed the model code and performed the
simulations. Benya Wang prepared the paper with contributions from all coauthors.

*Competing interests.* The authors declare that they have no conflict of interest.

*Acknowledgements.* The authors acknowledge Peisheng Huang and Brendan Busch for providing the historical nutrient data.

*Financial support.* Benya Wang was supported by a postgraduate scholarship provided by the CRC for Water Sensitive Cities. Matthew R. Hipsey received funding support from Australian Research Council project LP150100451.

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





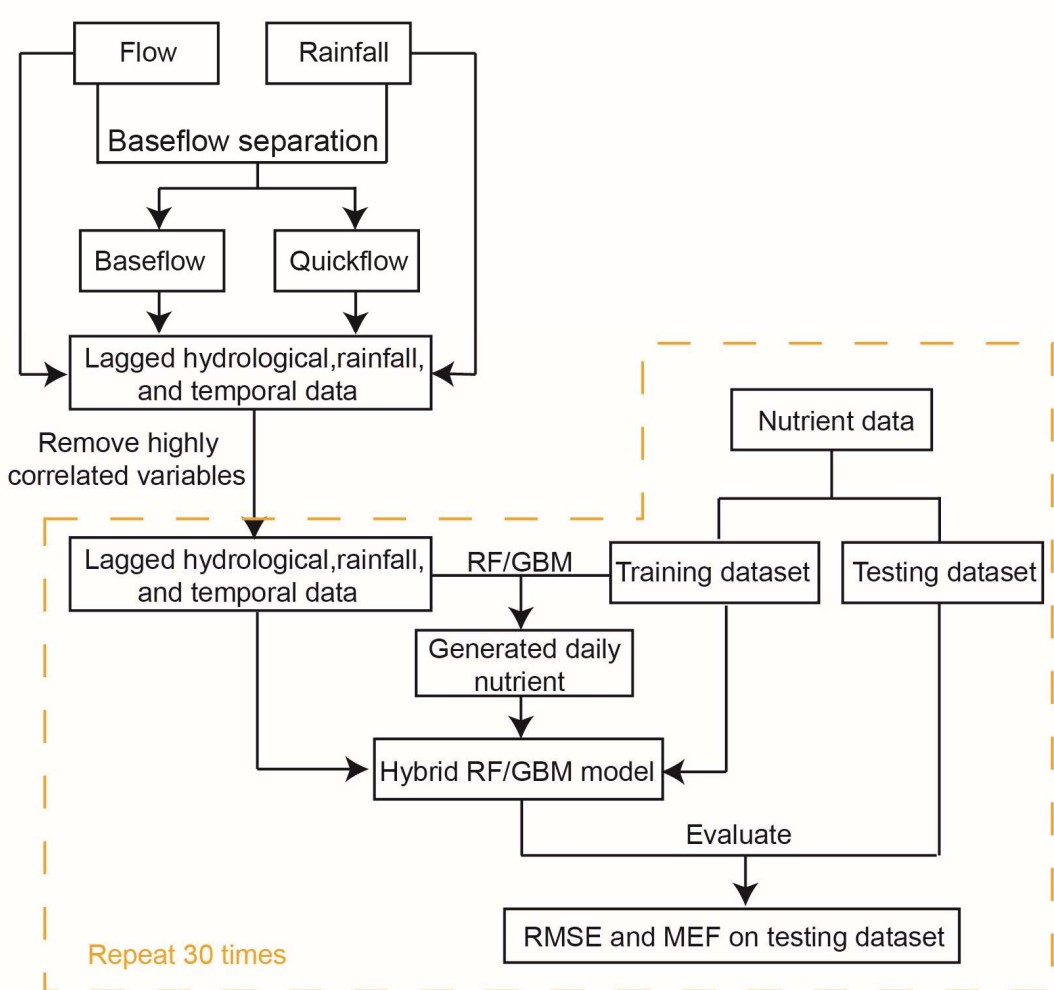

**Figure 1. Overall modelling processes of ML-SWAN.**




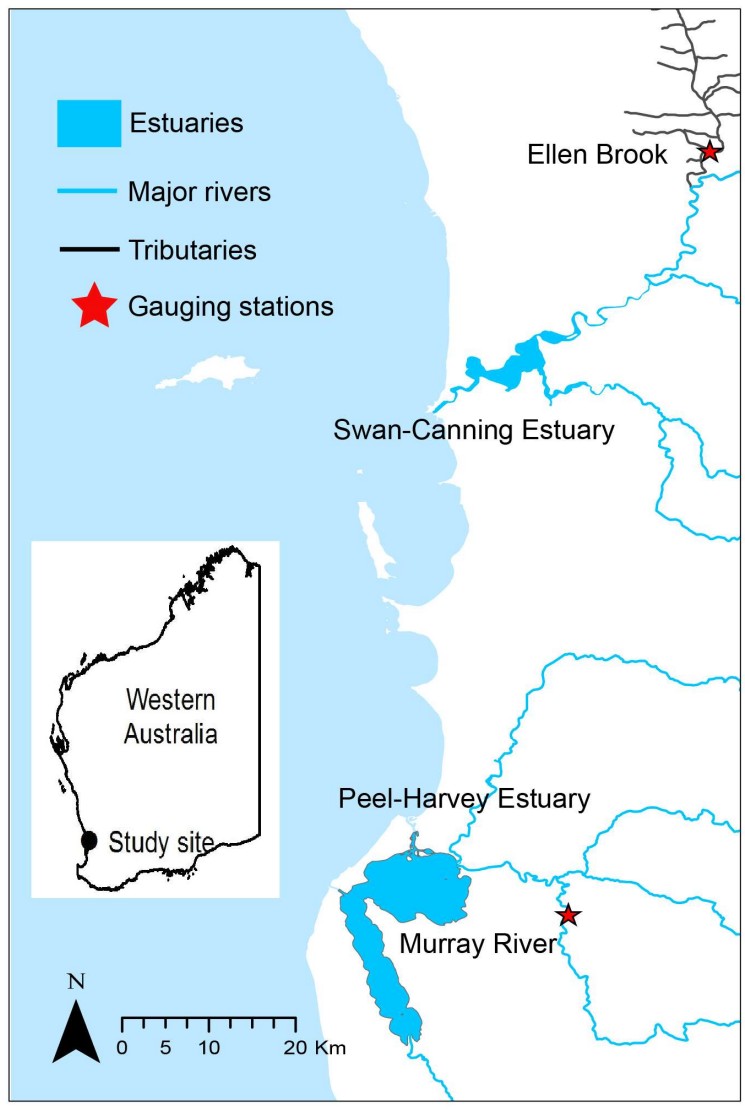

**Figure 2. The location of Ellen Brook and Murray River.**



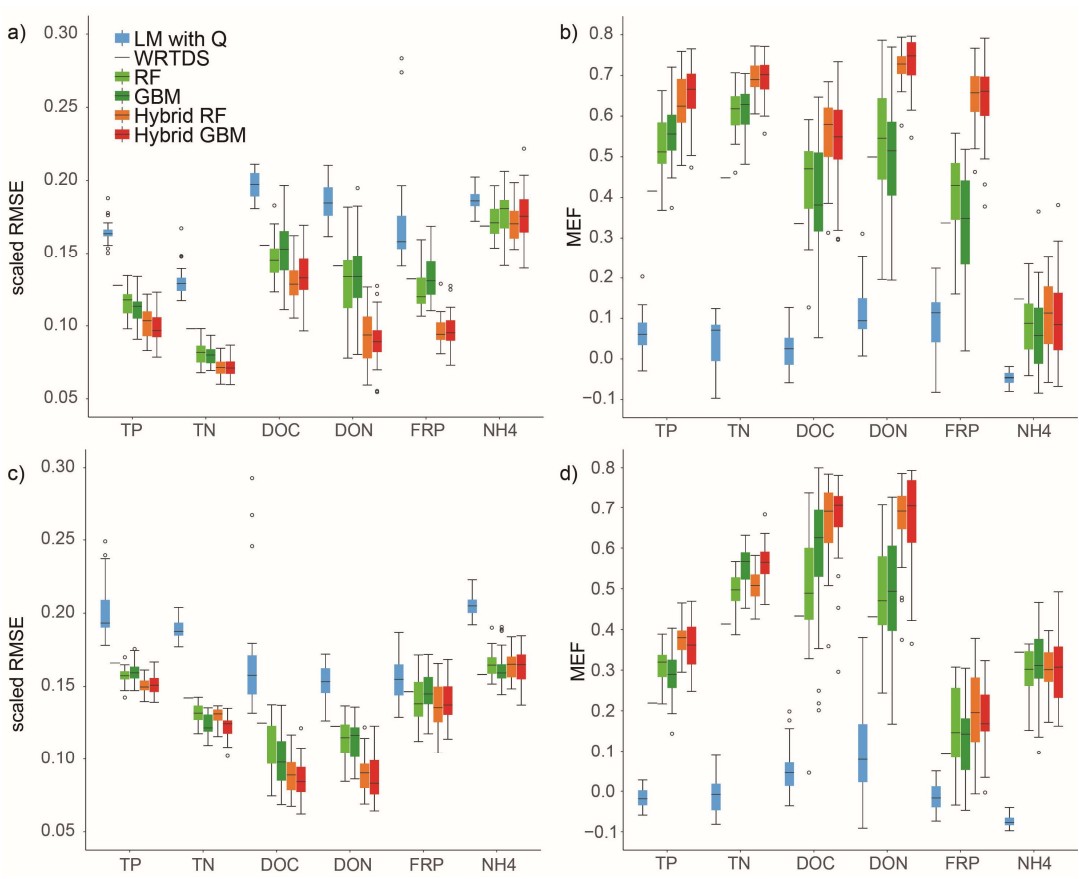

**Figure 3. Model performance across six nutrients and the two sites: a) RMSE and b) MEF for Ellen Brook; c) RMSE and d) MEF results for Murray River.**





Figure 4. Daily TN generated by the six models, for Ellen Brook.

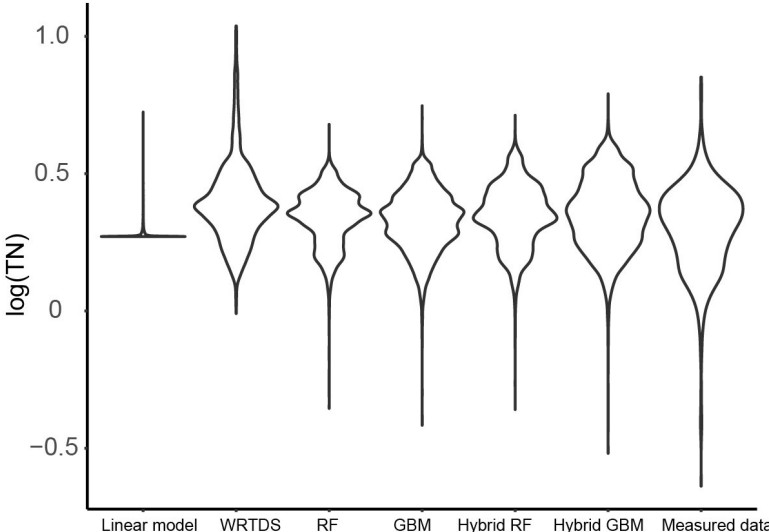

**Figure 5. The distribution of the daily TN generated by the six models, and of the measured TN data in Ellen Brook.**





**Figure 6. Variable importance in the hybrid GBM for TN prediction in a) Ellen Brook and b) Murray River.**



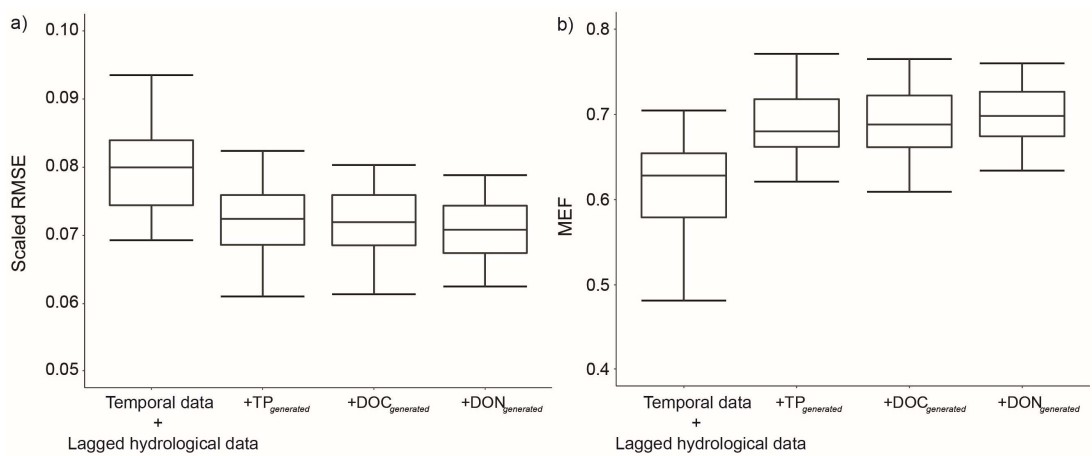

**Figure 7. Model performance for TN prediction across different input variables, for Ellen Brook.**

**Table 1. Variable list and descriptions.**

| Variable type | Variable name | Abbreviation | Unit | Data source |
|---|---|---|---|---|
| Hydrological data | total discharge | $Q$ | m³/s | data.wa.gov.au |
| | Average total discharge in last x days | $\bar{Q}_x$ | m³/s | lagged average |
| | quickflow | $QF$ | m³/s | equation 5 |
| | Average quickflow in last x days | $\overline{QF}_x$ | m³/s | lagged average |
| | baseflow | $BF$ | m³/s | equation 5 |
| | Average quickflow in last x days | $\overline{BF}_x$ | m³/s | lagged average |
| Temporal data | Julian day | $JD$ | | recorded |
| | cos (Julian day) | $Cos(JD)$ | | calculated |
| | sin (Julian day) | $Sin(JD)$ | | calculated |
| Metrological data | rainfall | $P$ | mm | www.bom.gov.au |
| | cumulated rainfall in last x days | $\sum_{1}^{x} P$ | mm | lagged sum |
| Nutrient data | total nitrogen | $TN$ | mg/L | wir.water.wa.gov.au |
| | total phosphorus | $TP$ | mg/L | wir.water.wa.gov.au |
| | dissolved organic carbon | $DOC$ | mg/L | wir.water.wa.gov.au |
| | dissolved organic nitrogen | $DON$ | mg/L | wir.water.wa.gov.au |

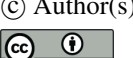



| | ammonia | $NH_3$ | mg/L | wir.water.wa.gov.au |
|---|---|---|---|---|
| | filterable reactive phosphorus | $FRP$ | mg/L | wir.water.wa.gov.au |
| | generated dissolved organic nitrogen | $DON_{generated}$ | mg/L | generated by the intermediate model |
| | generated total phosphorus | $TP_{generated}$ | mg/L | generated by the intermediate model |
| | generated dissolved organic carbon | $DOC_{generated}$ | mg/L | generated by the intermediate model |

**Table 2. Hydrological characteristics of the two tributaries.**

| Site | Hydrological type | Annual flow (GL) | Area (km$^2$) | Land use |
|---|---|---|---|---|
| Ellen Brook | ephemeral | 26.7 | 716 | rural, agricultural, and grazing |
| Murray River | perennial | 360 | 7855 | agricultural and natural reserves |

**Table 3. Nutrient sampling time and sample size in Ellen Brook and Murray River**

| Site | Nutrient | First measurement | Sample size |
|---|---|---|---|
| Ellen Brook | TN | 1990 | 1057 |
| | TP | 1990 | 1022 |
| | DOC | 1995 | 297 |
| | DON | 2006 | 129 |
| | FRP | 1990 | 404 |
| | $NH_3$ | 1990 | 356 |
| Murray River | TN | 1983 | 1648 |
| | TP | 1983 | 1662 |
| | DOC | 2006 | 209 |
| | DON | 2006 | 207 |
| | FRP | 1990 | 300 |
| | $NH_3$ | 1983 | 570 |
