# Peer review of "ML-SWAN-v1: a hybrid machine learning framework for the concentration prediction and discovery of transport pathways of surface water nutrients"

_Geoscientific Model Development, 2020_

## Referee Comment (RC1) · Thu Huong Thi Hoang (Referee) · 28 May 2020

Dear editor Please find attached review comment according to reviewing criteria. The manuscript is well written. However some discussion on ecological importances should be included to enhance pratical contribution of the research. I do hope that my comment can help to improve quality of this manuscript in particular and quality of the GMD in general.

Sincerely yours. Thu-Huong T. Hoang Hanoi University of Science and Technology

1. Does the paper address relevant scientific modelling questions within the scope of

[Figure]

GMD? Does the paper present a model, advances in modelling science, or a modelling protocol that is suitable for addressing relevant scientific questions within the scope of EGU?

The paper demonstrated a hybrid model of RM and GBM, which was able to predict and explain accurately the historical missing data as well as different pathways of TN export from two distinct catchments. The scientific question is suitable for the aim of GMD.

2. Does the paper present novel concepts, ideas, tools, or data?

In the present paper, a novel hybrid model has been developed and presented by the authors.

3. Does the paper represent a sufficiently substantial advance in modelling science?

The novel model developed by the authors could also be applied in different research areas, demonstrating a huge significance of this paper.

4. Are the methods and assumptions valid and clearly outlined?

Methods and data analysis were thoroughly presented.

5. Are the results sufficient to support the interpretations and conclusions?

Overall, the results and discussion satisfied the major aim of this paper, though several results were not carefully presented. For instance, the RMSE of results in Figure 4 and the meaning of Figure 5 in contribution to the comparison of models that the authors could pay more attention to.

6. Is the description sufficiently complete and precise to allow their reproduction by fellow scientists (traceability of results)? In the case of model description papers, it should in theory be possible for an independent scientist to construct a model that, while not necessarily numerically identical, will produce scientifically equivalent results. Model development papers should be similarly reproducible. For MIP and benchmarking pa-

pers, it should be possible for the protocol to be precisely reproduced for an independent model. Descriptions of numerical advances should be precisely reproducible.

The method as well as code and data provided by the authors could be utilized to reproduce a similar work.

7. Do the authors give proper credit to related work and clearly indicate their own new/original contribution?

The novelty was shown in comparison with a wide range of previous reports.

8. Does the title clearly reflect the contents of the paper? The model name and number should be included in papers that deal with only one model.

In the reviewer's point of view, the title could be improved to be more strength using the result of the discovery of pathways contribution of nutrient, not only prediction of concentration as its current state. The model name and version were provided.

9. Does the abstract provide a concise and complete summary?

The content of the abstract is totally good, however, it may better if the authors reduce the introduction of models and add more results of their works.

10. Is the overall presentation well structured and clear?

The paper was well and logically organized.

11. Is the language fluent and precise?

The authors used language precisely with clear meaning.

12. Are mathematical formulae, symbols, abbreviations, and units correctly defined and used?

Yes, it was accurately shown.

13. Should any parts of the paper (text, formulae, figures, tables) be clarified, reduced,

combined, or eliminated?

The manuscript more focuses on modelling techniques, only a few ecological discussion was provided. The manuscript provided some discussion on the source of TN in Ellen Brook and Murray River, however, the discussion should be presented better to avoid subjective idea only reflect author assumption. Discussion should better follow results and references The main idea of the Ecological Modelling is not only a prediction tool but also an explanation of ecological significance and pattern of environmental variables. The paper will be greatly improved if the authors spent more discussion on temporal and spatial patterns of predicted variables. Main question can be - How different b/w patterns of DON, TN, NH-N. How results can be used to explain the source of nutrient, - Transformation of nitrogen (in different forms of NH4-N, TN, DON, etc.) from source to river water bodies. - Solution to improve eutrophication situation in river

14. Are the number and quality of references appropriate?

Yes.

15. Is the amount and quality of supplementary material appropriate? For model description papers, authors are strongly encouraged to submit supplementary material containing the model code and a user manual. For development, technical, and benchmarking papers, the submission of code to perform calculations described in the text is strongly encouraged.

The authors provided sufficiently the code and data of the developed model.

---

## Referee Comment (RC2) · Anonymous Referee #2 · 2 Jun 2020

The manuscript presents a useful approach to simulate high temporal resolution water quality data required as inputs to receiving water modelling. In my opinion the work is publication worthy and generally well written.

The hybrid approach is novel as far as I'm aware, to predict a one water quality constituent for the current time step, as an input to predict a different constituent. The results demonstrate that this improves model performance compared to predicting the target constituent directly. This suggests that the different water quality parameters are highly correlated, and provide additional information that is not contained within the other inputs (of baseflow, quickflow, rainfall, day of the year and lags or averages of

these). Two additions are recommended related to this observation: 1. Presentation of the relationships between in the inputs and outputs would be informative. As noted in the discussion (line 445) correlation tests can be used to assess the which parameters would be useful for pre-generation. Potentially a scatter plot matrix, with correlation values on the lower diagonal, would be useful to show the relationships concisely. 2. It would be useful for the authors to speculate in the discussion, or even undertake the modelling, if training the model on the observed values for the hybrid model, rather than the generated values, would further improve performance. Possibly this would allow the data driven model to fit the underlying relationships more accurately, even if when implemented less accurate predicted values must be used as the inputs. Or possibly the hybrid model is correcting for some of the errors in the generated inputs if there is a systematic bias, and this would degrade performance.

Line 200: Please provide further information on how the 80:20 data split was implemented. Was the last 20% of the time series used for testing, or was a more complex method used? If this was the approach, does this explain why the DON, and DOC for Murray River, was the most important input, because this data is only present in the testing period? As noted in the manuscript, there appears to be a long term trend in the dataset, and only using this data that is more representative of the testing dataset may improve performance? Possibly data from the 1990s does not represent the responses experienced in the more recent testing period? The authors should further comment on these temporal and data length issues.

Line 316: claims that the generated data enables the hybrid ML to capture long-term trends. Please elaborate on how this is possible, as it is not clear to me. A given set of inputs to the stand alone ML to predict the generated data will give the same result in 1990 as 2018, so how can long term trends be captured? As outlined above, possibly because using inputs only collected in the more recent period is more representative of the testing data?

Section 4.2 moves from considering all nutrients to TN only. Please add a justification

why this was the case. Is this because TN is the more relevant for the application, or for the sake of brevity only one parameter is analysed in more detail.

Analysis of the relative importance of the different inputs is a valuable addition to the manuscript, which is often not considered in machine learning work. Overall, I consider this work of publishable merit subject to these minor additions and clarifications.

Minor points The first sentence of the abstract would benefit from being rewritten. Suggest making this two sentences and reworking, e.g. nutrient data is monitoring, not necessary for monitoring, for example.

I found the term "temporal data" confusing, I was assuming this meant lagged data, Qb,t-1, for example. It wasn't until I found Table 1 that this means Julian day of the year. It is suggested to be more specific, and call this "Julian day" or "seasonal component" or similar.

Line 70: Another relevant study for hybrid modelling may be Hunter et al. (2018) https://doi.org/10.5194/hess-22-2987-2018

Line 164: remove are from GBM model are generally have less. . .

Line 214: . . .. Can be divided into there stages, add s to stages.

Line 268: I was confused by this sentence, in that lower RMSE and increased MEF are both improved performance, but opposite patterns. Suggest "overall, the scaled RMSE improved from LM. . .. And the same pattern was found form MEF. . .

Line 277: suggest add to the end of this paragraph, "as such, the WRTDS results are not directly comparable to the other methods"

Line 389: the temporal data was more useful for the perennial catchment, Murray River. Could this be because seasonal information is captured in other inputs, e.g. Q for the more ephemeral catchment?

Line 399: Please elaborate further on the method used in this section, it is not clear

what has changed. Should "gradually" be "sequentially"?

---

## Author Comment (AC1) · 5 Jun 2020

5. Are the results sufficient to support the interpretations and conclusions? Overall, the results and discussion satisfied the major aim of this paper, though several results were not carefully presented. For instance, the RMSE of results in Figure 4 and the meaning of Figure 5 in contribution to the comparison of models that the authors could pay more attention to. Response: Thanks for your comments. We will add more content about Figure 4 and 5 results in the revised manuscript.

8. Does the title clearly reflect the contents of the paper? The model name and number should be included in papers that deal with only one model. In the reviewer's point of

view, the title could be improved to be more strength using the result of the discovery of pathways contribution of nutrient, not only prediction of concentration as its current state. The model name and version were provided.

9. Does the abstract provide a concise and complete summary? The content of the abstract is totally good, however, it may better if the authors reduce the introduction of models and add more results of their works. Response to comment 8 and 9: The title and the abstract would be updated to include more result information.

13. Should any parts of the paper (text, formulae, figures, tables) be clarified, reduced, combined, or eliminated? The manuscript more focuses on modelling techniques, only a few ecological discussion was provided. The manuscript provided some discussion on the source of TN in Ellen Brook and Murray River, however, the discussion should be presented better to avoid subjective idea only reflect author assumption. Discussion should better follow results and references The main idea of the Ecological Modelling is not only a prediction tool but also an explanation of ecological significance and pattern of environmental variables. The paper will be greatly improved if the authors spent more discussion on temporal and spatial patterns of predicted variables. Main question can be - How different b/w patterns of DON, TN, NH-N. How results can be used to explain the source of nutrient, - Transformation of nitrogen (in different forms of NH4-N, TN, DON, etc.) from source to river water bodies. - Solution to improve eutrophication situation in river. Response: Thanks for the comments. It is a really good idea to add these discussion points. We will add another section to discuss how different patterns of nutrients indicate nutrient sources and nitrogen transformation in the revised manuscript.

---

## Author Comment (AC2) · 6 Jun 2020

Two additions are recommended related to this observation: 1. Presentation of the relationships between in the inputs and outputs would be informative. As noted in the discussion (line 445) correlation tests can be used to assess the which parameters would be useful for pre-generation. Potentially a scatter plot matrix, with correlation values on the lower diagonal, would be useful to show the relationships concisely.

Response: Thanks for your comments. Please find the attached figure. We did this analysis before the modelling phase. There are strong relationships between nutrients. This is also one of the assumptions for this hybrid model that if we can pre-generate

these nutrients and they can be further used in the final model to predict final target nutrient. In that case, the model could have higher accuracy. We will update the manuscript to include this figure in the result section or in the supplement.

2. It would be useful for the authors to speculate in the discussion, or even undertake the modelling, if training the model on the observed values for the hybrid model, rather than the generated values, would further improve performance. Possibly this would allow the data driven model to fit the underlying relationships more accurately, even if when implemented less accurate predicted values must be used as the inputs. Or possibly the hybrid model is correcting for some of the errors in the generated inputs if there is a systematic bias, and this would degrade performance.

Response: Thanks for the suggestion. However, there is only very limited number of data (less than 30 samples) that have completed nutrient species to do the modelling. The hybrid model should have higher accuracy using observed values instead of the pre-generated data. We will add this in the discussion.

Line 200: Please provide further information on how the 80:20 data split was implemented. Was the last 20% of the time series used for testing, or was a more complex method used? If this was the approach, does this explain why the DON, and DOC for Murray River, was the most important input, because this data is only present in the testing period? As noted in the manuscript, there appears to be a long term trend in the dataset, and only using this data that is more representative of the testing dataset may improve performance? Possibly data from the 1990s does not represent the responses experienced in the more recent testing period? The authors should further comment on these temporal and data length issues.

Response: The main aim of this research is to rebuild the historical nutrient data and explore the short- and long-term changes. The first step is to verify the model performance. In that case, we randomly divided data into 80:20, built the model, tested on testing data (20%), and then repeated all steps for 30 times to further test model

uncertainty and stability (Figure 3). After this, all data points including the testing data were then used to rebuild the historical nutrient data (Figure 4). The different feature importance of DOC and DON in Murray River and Ellen Brook may due to the different nutrient sources and water pathways. We think it is a really good idea to compare model performance on more recent data and old data but it may out of this paper's scope.

Line 316: claims that the generated data enables the hybrid ML to capture long-term trends. Please elaborate on how this is possible, as it is not clear to me. A given set of inputs to the stand-alone ML to predict the generated data will give the same result in 1990 as 2018, so how can long term trends be captured? As outlined above, possibly because using inputs only collected in the more recent period is more representative of the testing data?

Response: The stand-alone model and the hybrid model used the same dataset to build and test model performance. If the pattern only exists in the recent samples, then both stand-alone and hybrid model should have similar fluctuation. The pre-generated nutrient is the only difference between the stand-alone model and the hybrid model. If there are long-term trends in the nutrient concentrations (e.g., TN), similar trends should also exist in the components of TN (either DON or DIN). The pre-generated nutrients emphasise this impact. That is why we suggested the pre-generated data in the hybrid model helped the model to capture long-term trends.

Section 4.2 moves from considering all nutrients to TN only. Please add a justification why this was the case. Is this because TN is the more relevant for the application, or for the sake of brevity only one parameter is analysed in more detail.

Response: Thanks for the comment. TN was selected because TN is the most important and most frequently measured nutrient in many places. In addition, we want this paper to be more concise. That is why only TN was analysed in detail. This hybrid method can be used for other nutrients

Minor points The first sentence of the abstract would benefit from being rewritten. Suggest making this two sentences and reworking, e.g. nutrient data is monitoring, not necessary for monitoring, for example. I found the term "temporal data" confusing, I was assuming this meant lagged data, Qb,t-1, for example. It wasn't until I found Table 1 that this means Julian day of the year. It is suggested to be more specific, and call this "Julian day" or "seasonal component" or similar. Line 70: Another relevant study for hybrid modelling may be Hunter et al. (2018) https://doi.org/10.5194/hess-22-2987-2018 Line 164: remove are from GBM model are generally have less. . . Line 214: . . .. Can be divided into there stages, add s to stages. Line 268: I was confused by this sentence, in that lower RMSE and increased MEF are both improved performance, but opposite patterns. Suggest "overall, the scaled RMSE improved from LM. . .. And the same pattern was found form MEF. . . Line 277: suggest add to the end of this paragraph, "as such, the WRTDS results are not directly comparable to the other methods" Line 389: the temporal data was more useful for the perennial catchment, Murray River. Could this be because seasonal information is captured in other inputs, e.g. Q for the more ephemeral catchment? Line 399: Please elaborate further on the method used in this section, it is not clear what has changed. Should "gradually" be "sequentially"?

Response: Thanks for these detailed comments. We will update the manuscript accordingly.
* * *
[Figure]

**Fig. 1.** correlation_between_nutrients

---

## Author Response (AR1)

*Faculty of Engineering and Mathematical Sciences*
*School of Civil, Environmental and Mining Engineering*
*The University of Western Australia*
*M015*
*35 Stirling Highway*
*Perth, Western Australia 6009*
*AUSTRALIA*

18 July 2020

Responses to reviewers' comments

We thank the reviewers for their detailed and excellent comments. In response to the comments, we have made substantial changes to the manuscript and believe that it is much improved. Specifically,

- The title and the abstract have been rewritten
- The discussion section was extended with more content of modelling results.
- A new figure was added to the main manuscript and several figures were updated.
- A new figure was added to the supplement

We have provided below our detailed responses to individual comments from the reviewers.

Please don't hesitate to contact me if you require any clarifications.

Yours sincerely

Benya Wang

Comments from reviewer #1 (Thu Huong Thi Hoang):

45

1. Does the paper address relevant scientific modelling questions within the scope of GMD? Does the paper present a model, advances in modelling science, or a modelling protocol that is suitable for addressing relevant scientific questions within the scope of EGU?

*Comments:* The paper demonstrated a hybrid model of RM and GBM, which was able to predict and explain

50      accurately the historical missing data as well as different pathways of TN export from two distinct catchments. The scientific question is suitable for the aim of GMD.

*Response:* Thanks for the comment.

2. Does the paper present novel concepts, ideas, tools, or data?

55      *Comments:* In the present paper, a novel hybrid model has been developed and presented by the authors.

*Response:* Thanks for the comment.

3. Does the paper represent a sufficiently substantial advance in modelling science?

*Comments:* The novel model developed by the authors could also be applied in different research areas, demonstrating

60      a huge significance of this paper.

*Response:* Thanks for the comment.

4. Are the methods and assumptions valid and clearly outlined?

*Comments:* Methods and data analysis were thoroughly presented.

65      *Response:* Thanks for the comment.

5. Are the results sufficient to support the interpretations and conclusions?

*Comments:* Overall, the results and discussion satisfied the major aim of this paper, though several results were not carefully presented. For instance, the RMSE of results in Figure 4 and the meaning of Figure 5 in contribution to the

70      comparison of models that the authors could pay more attention to.

*Response*: Thanks for your comments. More content about Figure 4 and 5 results were added to the revised manuscript.

*Changes in manuscript:* Several paragraphs were updated in section 4.2 and please find the update content in the revised manuscript.

75

6. Is the description sufficiently complete and precise to allow their reproduction by fellow scientists (traceability of results)? In the case of model description papers, it should in theory be possible for an independent scientist to construct a model that, while not necessarily numerically identical, will produce scientifically equivalent results. Model development papers should be similarly reproducible. For MIP and benchmarking pa-pers, it should be possible for the protocol to be precisely reproduced
80    for an independent model. Descriptions of numerical advances should be precisely reproducible.

*Comments:* The method as well as code and data provided by the authors could be utilized to reproduce a similar work.

*Response:* Thanks for the comment.

85   7. Do the authors give proper credit to related work and clearly indicate their own new/original contribution?

*Comments:* The novelty was shown in comparison with a wide range of previous reports.

*Response:* Thanks for the comment.

8. Does the title clearly reflect the contents of the paper? The model name and number should be included in papers that deal
90    with only one model.

*Comments:* In the reviewer's point of view, the title could be improved to be more strength using the result of the discovery of pathways contribution of nutrient, not only prediction of concentration as its current state. The model name and version were provided.

*Response:* Thanks for the comment. The title is updated to include more result information.

95    *Changes in manuscript:* the title has been changed as "ML-SWAN-v1: a hybrid machine learning framework for the concentration prediction and discovery of transport pathways of surface water nutrients"

9. Does the abstract provide a concise and complete summary?

*Comments:* The content of the abstract is totally good, however, it may better if the authors reduce the introduction
100   of models and add more results of their works.

*Response:* Thanks for the comment. The abstract is updated to include more result information.

*Changes in manuscript:* please find the updated abstract below.

Nutrient data from catchments discharging to receiving waters is monitoring for catchment management. However, nutrient data are often sparse in time and space and have non-linear responses to environmental factors, making it
105   difficult to systematically analyse long- and short-term trends and undertake nutrient budgets. To address these challenges, we developed a hybrid machine learning (ML) framework that first separated baseflow and quickflow from total flow, generated data for missing nutrient species, and included pre-generated nutrient data as additional variables in a final simulation of tributary water quality. Hybrid random forest (RF) and gradient boosting machines

(GBM) models were employed and their performance compared with a linear model, a multivariate weighted regression model and stand-alone RF and GBM models that did not pre-generate nutrient data. The six models were used to predict six different nutrients discharged from two study sites in Western Australia: Ellen Brook (small and ephemeral) and the Murray River (large and perennial). Our results showed that the hybrid RF and GBM models had significantly higher accuracy and lower prediction uncertainty for almost all nutrient species across the two sites. The pre-generated nutrient and hydrological data were highlighted as the most important components of the hybrid model. The model results also indicated different hydrological transport pathways for TN export from the two tributary catchments. We demonstrated that the hybrid model provides a flexible method to combine data of varied resolution and quality, and is accurate for the prediction of responses of surface water nutrient concentrations to hydrologic variability.

10. Is the overall presentation well structured and clear?

*Comments:* The paper was well and logically organized.

*Response:* Thanks for the comment.

11. Is the language fluent and precise?

*Comments:* The authors used language precisely with clear meaning.

*Response:* Thanks for the comment.

12. Are mathematical formulae, symbols, abbreviations, and units correctly defined and used?

*Comments:* Yes, it was accurately shown.

*Response:* Thanks for the comment.

13. Should any parts of the paper (text, formulae, figures, tables) be clarified, reduced, combined, or eliminated?

*Comments:* The manuscript more focuses on modelling techniques, only a few ecological discussion was provided. The manuscript provided some discussion on the source of TN in Ellen Brook and Murray River, however, the discussion should be presented better to avoid subjective idea only reflect author assumption. Discussion should better follow results and references The main idea of the Ecological Modelling is not only a prediction tool but also an explanation of ecological significance and pattern of environmental variables. The paper will be greatly improved if the authors spent more discussion on temporal and spatial patterns of predicted variables. Main question can be - How different b/w patterns of DON, TN, NH-N. How results can be used to explain the source of nutrient, - Transformation of nitrogen (in different forms of NH4-N, TN, DON, etc.) from source to river water bodies. - Solution to improve eutrophication situation in river

*Response*: Thanks for the comment. It is exactly right that the purpose of environmental modelling is to have a better understanding of the ecosystem. The main purpose of this paper is to introduce the hybrid model and that's why we didn't put main focus on the application of this method before. In the updated manuscript, more contents about the

145        sources of TN in Ellen Brook was added to section 5.1.

*Changes in manuscript:* Several paragraphs were updated. Please directly refer to section 5.1 for the changes.

14. Are the number and quality of references appropriate?

*Comments:* Yes.

150        *Response:* Thanks for the comment.

15. Is the amount and quality of supplementary material appropriate? For model description papers, authors are strongly encouraged to submit supplementary material containing the model code and a user manual. For development, technical, and benchmarking papers, the submission of code to perform calculations described in the text is strongly encouraged.

155        *Comments:* The authors provided sufficiently the code and data of the developed model

*Response:* Thanks for the comment.

Comments from reviewer #2:

160

*Comments 1:* Presentation of the relationships between in the inputs and outputs would be informative. As noted in the discussion (line 445) correlation tests can be used to assess the which parameters would be useful for pre-generation. Potentially a scatter plot matrix, with correlation values on the lower diagonal, would be useful to show the relationships concisely.

165        *Response*: Thanks for your comments. We did this analysis before the modelling phase. There are strong relationships between nutrients (see Figure A.1). This is also one of the assumptions for this hybrid model that if we can pre-generate these nutrients and they can be further used in the final model to predict final target nutrient. In that case, model could have higher accuracy.

*Changes in manuscript:* please find the Figure A.2 1 in the supplement document.

170

*Comments 2:* It would be useful for the authors to speculate in the discussion, or even undertake the modelling, if training the model on the observed values for the hybrid model, rather than the generated values, would further improve performance. Possibly this would allow the data driven model to fit the underlying relationships more accurately, even if when implemented less accurate predicted values must be used as the inputs. Or possibly the hybrid model is correcting for some of the errors in

175        the generated inputs if there is a systematic bias, and this would degrade performance.

*Response*: Thanks for the suggestion. The hybrid model should have higher accuracy using observed values instead of the pre-generated data. However, there are only very limited number of data (less than 30 samples) that have completed nutrient species to do this test.

*Changes in manuscript:* no changes in the manuscript.

180

*Comments 3 (Line 200):* Please provide further information on how the 80:20 data split was implemented. Was the last 20% of the time series used for testing, or was a more complex method used? If this was the approach, does this explain why the DON, and DOC for Murray River, was the most important input, because this data is only present in the testing period? As noted in the manuscript, there appears to be a long term trend in the dataset, and only using this data that is more representative of the testing dataset may improve performance? Possibly data from the 1990s does not represent the responses experienced in the more recent testing period? The authors should further comment on these temporal and data length issues.

*Response*: The main aim of this research is to test the hybrid model, rebuild the historical nutrient data, and explore the short- and long-term nutrient changes. The first step is verifying the model performance. In that case, we randomly divided data into 80:20, built the model, tested on testing data (20%), and then repeated all steps for 30 times to further test model uncertainty and stability (Figure 3). After this, all data points including the testing data were then used to rebuild the historical nutrient data (Figure 4). The different feature importance of DOC and DON in Murray River and Ellen Brook may due to the different nutrient sources and water pathways. We think it is a really good idea to compare model performance on more recent data and old data but it may out of this paper's scope.

*Changes in manuscript:* The main aim of this research is to test the hybrid model, rebuild the historical nutrient data, and explore the short- and long-term nutrient changes. The first step is verifying the model performance. In that case, the data were randomly divided into 80:20. Different models were built and tuned on the training dataset (80%); the testing dataset (20%) was saved for the final test. To further test model uncertainty and stability, the divided and tested processes were repeated 30 times except WRTDS. After this, all data points including the testing data were then used to rebuild the historical nutrient data.

200

*Comments 4 (Line 316):* claims that the generated data enables the hybrid ML to capture long-term trends. Please elaborate on how this is possible, as it is not clear to me. A given set of inputs to the stand alone ML to predict the generated data will give the same result in 1990 as 2018, so how can long term trends be captured? As outlined above, possibly because using inputs only collected in the more recent period is more representative of the testing data?

205

*Response*: The stand-alone model and the hybrid model used same dataset to build and test model performance. If the pattern only exists in the recent samples, then both stand-alone and hybrid model should have similar fluctuation. The pre-generated nutrient is the only difference between stand-alone model and hybrid model. If there are long-term trends in the nutrient concentrations (e.g., TN), similar trends should also exist in the components of TN (either DON

or DIN). The pre-generated nutrients emphasise this impact. That is why we suggested the pre-generated data in the hybrid model helped model to capture long-term trends.

*Changes in manuscript:* The pre-generated nutrient is the only difference between stand-alone model and hybrid model. If there are long-term trends in nutrient concentrations (e.g., TN), similar trends should also exist in the components of TN (either DON or dissolved inorganic nitrogen). The pre-generated nutrients emphasise this impact on the hybrid model. This suggests that the generated nutrient data could provide additional information that allowed the hybrid ML to capture long-term trends; this information was not included in the seasonal components, but existed in the generated nutrient data.

*Comments 5:* Section 4.2 moves from considering all nutrients to TN only. Please add a justification why this was the case. Is this because TN is the more relevant for the application, or for the sake of brevity only one parameter is analysed in more detail.

*Response:* Thanks for the comment. TN was selected because TN is the most important and most frequently measured nutrient in many places. In addition, we want this paper to be more concise. That is why only TN was analysed in detail. This hybrid method can be used for other nutrients.

*Changes in manuscript:* Model performance for six nutrients was compared in last section. To make this section more concise, these six models were then compared in their ability to generate daily TN in Ellen Brook from 01/01/1989 to 16/07/2018 (Figure 6Figure 7). TN was selected because TN is the most important and most frequently measured nutrient in many places. This hybrid method can also be used for other nutrients (see results in the supplement document).

*Comments 6:* Analysis of the relative importance of the different inputs is a valuable addition to the manuscript, which is often not considered in machine learning work. Overall, I consider this work of publishable merit subject to these minor additions and clarifications.

*Response:* Thanks for the comment.

Minor points

*Comments 7:* The first sentence of the abstract would benefit from being rewritten. Suggest making this two sentences and reworking, e.g. nutrient data is monitoring, not necessary for monitoring, for example. I found the term "temporal data" confusing, I was assuming this meant lagged data, Qb,t-1, for example. It wasn't until I found Table 1 that this means Julian day of the year. It is suggested to be more specific, and call this "Julian day" or "seasonal component" or similar.

*Response:* thanks for the comment. This sentence has been rewritten and separated as two sentences. All "temporal data" has been changed as "seasonal component".

*Changes in manuscript:* Nutrient data from catchments discharging to receiving waters is monitoring for catchment management. However, nutrient data are often sparse in time and space and have non-linear responses to environmental factors, making it difficult to systematically analyse long- and short-term trends and undertake nutrient budgets.

*Comments 8 (Line 70):* Another relevant study for hybrid modelling may be Hunter et al. (2018) https://doi.org/10.5194/hess-22-2987-2018

*Response:* thanks for sharing this paper. It's interesting to see that both this study and our study found the hybrid model could achieve the best performance than stand-along machine learning model and simple process-based model. We have included this paper in the discussion section.

*Changes in manuscript:* the following content was added.

Similar results were also found in Hunter et al. (2018) that a hybrid process-driven and ANN model was compared with the stand-alone ANN model and the process-driven model. In their study, the hybrid also achieved the best performance followed by stand-alone ANN. The process-driven benchmark model had significantly lower accuracy than other two models.

*Comments 9 (Line 164):* remove are from GBM model are generally have less …

*Response:* Thanks for the comment.

*Changes in manuscript:* this sentence has been removed from the manuscript.

*Comments 10 (Line 214):* Can be divided into there stages, add s to stages.

*Response:* Thanks for the comment.

*Changes in manuscript:* The overall processes of ML-SWAN can be divided into three stages (Figure 1).

*Comments 11 (Line 268):* I was confused by this sentence, in that lower RMSE and increased MEF are both improved performance, but opposite patterns. Suggest "overall, the scaled RMSE improved from LM ... And the same pattern was found form MEF …

*Response:* Thanks for the comment.

*Changes in manuscript:* Overall, the scaled RMSE reduced from LM, WRTDS, stand-alone ML, and hybrid ML for all nutrients except NH3, and the same pattern was found for MEF in both Ellen Brook and Murray River (Figure 3).

*Comments 12 (Line 277):* suggest add to the end of this paragraph, "as such, the WRTDS results are not directly comparable to the other methods"

*Response:* Thanks for the comment.

*Changes in manuscript:* this sentence has been added to the end of this paragraph.

*Comments 13 (Line 389):* the temporal data was more useful for the perennial catchment, Murray River. Could this be because seasonal information is captured in other inputs, e.g. Q for the more ephemeral catchment?

280    *Response:* thanks for the comment. This could be a possible reason for the lower feature importance of seasonal components in Ellen Brook. But results in Figure 4-f and Figure A.2-f suggest that TN in Murray River has higher seasonal changes than Ellen Brook. Quickflow and baseflow data in Ellen Brook didn't exhibit significantly higher features importance than Murray River (Figure 6).

       *Changes in manuscript:* This may because seasonal information is captured in other inputs in Ellen Brook (e.g.,
285    quickflow and baseflow). But the main reason is the stronger seasonal TN signals in Murray River compared to Ellen Brook. This finding is supported by the generated daily TN data for Murray River (see results in supplement A.2).

*Comments 14 (Line 399):* Please elaborate further on the method used in this section, it is not clear what has changed. Should "gradually" be "sequentially"?

290    *Response:* Thanks for the comment. More contents were added to this section.

[revised manuscript text omitted]